# Dairy Product Intake and Cardiometabolic Diseases in Northern Sweden: A 33-Year Prospective Cohort Study

**DOI:** 10.3390/nu11020284

**Published:** 2019-01-28

**Authors:** Ingegerd Johansson, Anders Esberg, Lena M Nilsson, Jan-Håkan Jansson, Patrik Wennberg, Anna Winkvist

**Affiliations:** 1Department of Odontology, Umeå University, 90187 Umeå, Sweden; anders.esberg@umu.se; 2Department of Nutritional Research, Umeå University, 90187 Umeå, Sweden; lena.nilsson@umu.se (L.M.N.); anna.winkvist@nutrition.gu.se (A.W.); 3Department of Public Health and Clinical Medicine, Research Unit Skellefteå, Umeå University, 90187 Umeå, Sweden; janhakan.jansson@vll.se; 4Department of Public Health and Clinical Medicine, Family Medicine, Umeå University, 90187 Umeå, Sweden; patrik.wennberg@vll.se; 5Department of Internal Medicine and Clinical Nutrition, Sahlgrenska Academy, University of Gothenburg, 405 30 Gothenburg, Sweden

**Keywords:** dairy products, milk, cardiovascular disease, myocardial infarction, stroke, type 2 diabetes

## Abstract

Dairy products are important constituents of most diets, and their association with adverse health outcomes remains a focus. We characterized dairy food intake and examined associations with the incidence of type 2 diabetes (T2D), myocardial infarction (MI) or stroke among 108,065 Swedish men and women. Hazard ratios (HRs) and 95% CIs were estimated using the multivariable Cox proportional hazards models in a population characterized by high milk tolerance. During a mean follow-up of 14.2 years, 11,641 first-time events occurred. Non-fermented milk intake decreased, whereas butter intake increased over the period. For high intake of non-fermented milk, the HR (95% CI) for developing T2D and MI was 1.17 (1.03, 1.34) and 1.23 (1.10, 1.37), respectively, in men. A greater intake of butter, fermented milk, and cheese tended to be associated with a reduced risk of T2D and/or MI. Non-consumers and those who chose low-fat variants of the targeted dairy products had increased risk for T2D, MI, or stroke compared to those in the non-case group. Generally, effect-sizes were small. This prospective study found that non-fermented milk was associated with an increased risk for developing T2D and MI and that subjects abstaining from dairy products or choosing low-fat variants were at greater risk. However, the overall cardiometabolic risk of non-fermented milk intake was judged as low, since the effect sizes were small.

## 1. Introduction

Dairy products are a part of basic nutrition in most populations, which is reflected in both worldwide national dietary guidelines and concerns about the association of dairy consumption with health outcomes [1,2,3]. Milk, the basis for dairy products, is a complex product with an array of potentially health-associated components, such as innate immunity peptides, free and bound sugar structures, and various components for physiological processes [4]. Dairy products are also a major source of saturated fatty acids with assumed adverse effects on cardiometabolic events [3]. In many populations, milk consumption is constrained after childhood due to reduced lactase production [5]. In some populations, such as in the Scandinavian countries, Sweden, Denmark, and Norway, the ability to digest lactose has evolved, and milk consumption remains high into adulthood [5]. In 2013, Sweden had the fourth highest per capita milk consumption worldwide, with an average of 341.2 kg per person and year (butter excluded), corresponding to approximately 340 kCal/person per day according to FAO food balance sheets [6].

The associations between milk and dairy intake and cardiometabolic health outcomes have been assessed in cohort [7,8,9] and cohort meta-analysis [10,11,12,13,14,15,16] studies and in Mendelian randomization evaluations using SNPs of the MCM6 gene [17,18,19]. The results are inconsistent, with some protective associations [8,13,14,15,16], many null-associations [10,11,12], and some inverse association [9] when events, such as total and cause-specific mortality and incident development of various cardiovascular events, were targeted [17]. The reasons for this inconsistency are unclear, but potential factors include differences in population characteristics (e.g., levels and profiles of dairy intake), data precision, statistical approaches and whether Mendelian studies took population stratification into account [20].

The Northern Sweden Health and Disease Study (NSHDS) [21] is a longitudinal screening of adults that has been ongoing since 1986 in the northernmost counties of Sweden. Possibilities to link various Swedish population-based quality registers on mortality, morbidity, socio-economic factors, etc., allow for observational studies in a population characterized by significant cultural and genetic homogeneity. One characteristic of this population is the high prevalence of the lactase-persistence T allele (TT and CT in 92.4%) [22], with 4–10% of the population reported as lactose-intolerant [23]; in addition, this population was not only given general guidelines on reducing saturated fat intake [2] but also influenced by potential benefits from a high-fat/low-carbohydrate diet since approximately 2005 [24]. In concert, these conditions make the NSHDS stand out as complementary to the published milk and dairy association studies that are commonly performed in more heterogeneous and low-milk/dairy-consuming populations.

The aims of the present longitudinal cohort study were to characterize dairy food intake and its prospective association with incident cardiometabolic events, that is type 2 diabetes (T2D), myocardial infarction (MI) and stroke, in a large population with an evolutionarily high tolerance to milk.

## 2. Materials and Methods

### 2.1. Study Design and Study Participants

The Northern Sweden Diet Database (NSDD; http://www.biobank.umu.se/kostdatabasen/) contains information on respondents in the Northern Sweden Health and Disease Study (NSHDS, http://www.biobank.umu.se/nshds/) with at least one diet recording. In the NSHDS, data are collected within the Västerbotten Intervention Programme (VIP) and the Northern Sweden MONICA (Monitoring Trends and Determinants in Cardiovascular Disease) screenings as described previously [21]. Briefly, VIP continuously invites residents in the county of Västerbotten in Northern Sweden to a health examination at their nearest health centre when they turn 40, 50 and 60 years of age (for a period, 30-year-olds were also included). Close to 30% of the participants have a 10-year follow-up visit, and 7% have a 20-year follow-up visit. MONICA (http://www.org.umu.se/monica) invited a random population-based sample of 2000 to 2500 25- to 75-year-old subjects in the Västerbotten and Norrbotten counties in 1986, 1990, 1994, 1999, 2004, 2009, and 2014 to a health screening.

VIP and MONICA participants undergo an extensive health examination, including measurements of blood pressure, blood lipid profile and glucose levels before and after a glucose load, height and body weight, and they answer an extensive questionnaire on diet, lifestyle, health and life conditions using harmonized methods. The participation rate has varied over time with an average of 60%, and limited evidence of significant selection bias in relation to income, age and unemployment has been reported [25,26].

On 31 December 2016, the basic NSDD included 182,578 observations and 120,061 unique subjects (50.7% women). For the present study, observations were excluded if (i) the food intake recording was incomplete, i.e., ≥10% missing data and/or a missing portion indication, or included extreme (highest and lowest 1%) food intake levels (FILs) [27] or extreme energy intakes (lowest 1% and >5000 kCal); (ii) the height (<130 or >210 cm) or weight (<35 kg) values were implausible or BMI < 15.0 (in total 6960 subjects). Subjects who had immigrated or emigrated during the follow-up period were also excluded (in total 5036 subjects). The final study group was composed of 108,065 unique subjects (51.5% women) representing the first screening event between 1986 and 31 December 2016. The flow chart is shown in Figure 1.

### 2.2. Ethical Approval

The study protocol and data handling procedures were approved by the Regional Ethical Review Board of Northern Sweden, Umeå (Dnr 2013/332/31). The study was conducted in accordance with the Declaration of Helsinki and included the written informed consent of all study participants. The data generated and analyzed are not publicly available due to Swedish legislations but are available from the Department of Biobank Research, Umeå University, on reasonable request and with appropriate approvals. Information is available on the website (http://www.biobank.umu.se/).

### 2.3. Dietary Assessments

All participants completed a food frequency questionnaire (FFQ) targeting their dietary habits during the most recent year. Over the study period, two FFQ versions were used: a longer version (84 food items/aggregates) and a shortened version (64–66 food items/aggregates). The questions on dairy products, i.e., 3 questions for non-fermented milk (0.5%, 1.5% and 3.0% fat, respectively, referred to as low, medium and high fat milk), 2 questions for fermented milk (0.5 and ≥3% fat, respectively, referred to as low and high variants), 3 questions for butter (pure butter on bread or for cooking and one on a mixed spread with 70% butter), and 2 questions for cheese (10–15% fat and 28 or more % fat, and referred to as low or high variants), remained unchanged over the study period. The longer version, which was used until 1996, was completed by 36% of the participants, and the shorter version was completed by 64% of the participants at the participants’ first visit. Among the shorter FFQ versions, 15% had responded to the 64-question variant, 20% to the 65-question variant, and 65% to the 66-question variant. Intakes were reported on a fixed 9-level scale, ranging from never to 4 times or more per day. Meal-time portion sizes were aided by four colour pictures of plates with increasing amounts of staple foods (potato, rice, and/or pasta), main protein sources (meat and/or fish), and vegetables, and other foods, such as dairy products, had sex- and age-specific portion sizes or fixed sizes, e.g., an apple or egg [28]. The daily intake of energy and nutrients was estimated from food compositions provided by the National Food Agency of Sweden (www.livsmedelsverket.se/en/food-and-content/naringsamnen/livsmedelsdatabasen/). The relative validities were estimated against repeated 24-hour dietary records and biological markers, including a comparison between the long and short version [28,29,30,31].

### 2.4. Assessments of Myocardial Infarction, Stroke and Type 2 Diabetes

Information on all medical diagnoses or deaths was obtained by linking “Patient” and “Cause of death” registers at the National Board of Health and Welfare in Sweden (www.socialstyrelsen.se/register) using personal identification numbers. ICD-9 code 250 and ICD-10 codes E11.0–E11.9 were applied for T2D, ICD-9 code 410 and ICD-10 code I21 were applied for MI, and ICD-9 codes 430, 431, and 433–436 and ICD-10 codes I60, I61, I63 and I64 were applied for stroke. The first date of a registered T2D, MI, or stroke diagnosis was selected as the outcome and merged with NSDD/NSHDS data (Figure 1). For subjects with several incident cardiometabolic diagnoses, the first diagnosis was considered, and subjects with more than one outcome diagnosis at the first event were excluded from this study.

### 2.5. Assessment of a Non-Case Group

After exclusion of all subjects who (i) were diagnosed with T2D, MI/CHD, or stroke, (ii) had a CHD or stroke-related diagnosis and (iii) had died during the study period for reasons other than T2D, MI, or stroke, a non-case group of 86,931 subjects (52.9% women) remained (Figure 1).

### 2.6. Assessment of Metabolic Risk Markers

The BMI (body weight/height^2^) was calculated from body weight (kg) and height (m), measured when the participants were wearing light clothes and no shoes. Total cholesterol and triglycerides were analyzed in serum samples at health centres using a Reflotron benchtop analyzer (Boehringer Mannheim GmbH, Diagnostica, Germany) in the earliest years and an enzymatic routine method at the Clinical Chemistry Department at the nearest local hospital after 1 September 2009. Harmonisation was performed using an algorithm from a calibration set [32]. Blood glucose levels (b-glucose) were analyzed using a benchtop analyzer. Systolic and diastolic blood pressures were measured after 5 min of rest.

### 2.7. Assessment of Potential Confounding Lifestyle Factors

Information on tobacco use (smoking and Swedish snus (snuff)), highest level of education, physical activity in leisure time, whether a close relative had MI, stroke or T2D (yes/no), daily intakes of red meat, fruits and vegetables, and whole grain-containing foods, and total energy was collected from the questionnaires. Smoking and use of Swedish snus were categorized as never used, past daily or past occasional use, present daily or present occasional use. For education, participants were categorized into four levels with academic education as the highest level, and for physical activity, five levels reflected inactivity to having a training activity more than 3 times a week. Participants were classified into quintiles (by sex and 10-year age groups) based on the distribution of reported red meat, fruits and vegetables, whole grain-containing foods, and total energy intakes.

### 2.8. Data Processing and Statistical Analyses

Participants were categorized as non-consumers (those reporting null intake of a dairy food item) or consumers (those reporting any intake). After non-consumers had been placed in a “non-consumer” category, consumers were classified into quartiles based on the distribution of their reported intakes/day of each targeted dairy food item. Ranking was performed in sex and 10-year age strata. Thus, a total of five categories were compared.

Categorical measures are presented as proportions (%) and continuous measures as sex (if not split for sex)-, age-, and screening year-adjusted means with standard deviations (SDs) or 95% confidence limits (CIs). The differences between group numbers/means were not tested if the difference was interpreted as biologically non-meaningful and the large groups likely induced a statistical significance. When the tests were applied, the distributions of group numbers were tested with a chi-squared test and the differences between group means were tested with ANOVA, in general linear model multiple regression, with applicable covariates.

The 30-year time trend curves were searched for significant trend breaking points (joinpoints) using the Joinpoint Regression Program software (https://surveillance.cancer.gov/joinpoint/). This program identifies “joinpoints” in trend data by repeated permutation tests, and *p*-values are found by Monte Carlo methods [33].

Cox proportional hazards regression and hazard ratios (HRs) were calculated to estimate the risk of developing T2D, MI, or stroke during the follow-up period by dairy product category. The time (months) between the health screening and diagnosis or end of the study period (31 December 2016), whichever occurred first, was used as the time scale. The group with the lowest reported intake was the reference category. The proportional hazards assumption was assured by evaluating Schoenfeld residuals. Cox regression analyses were run both together and separately for men and women. Basic models included sex (if not split analyses), age, screening year and dairy product category. Adjusted models also included BMI, education level, physical activity in leisure time, smoking status, self-reported information on whether a close relative had MI, stroke or T2D, recruitment project (VIP or MONICA), and quintiles of red meat, fruit and vegetables, whole grains and energy intakes based on variables usually included as covariates in similar analyses [8]. To evaluate if/how the selection of reference category affected the hazard ratio (HR) scores, Cox regression analyses were also performed with the non-consumers as the reference category and the same covariates described above. Missing data were treated as a separate category.

The proportions of variance explained by dairy product intake (the effect sizes) for incident T2D, MI, or stroke were calculated as the eta squared in the GLM ANOVA in SPSS for sex and 10-year age strata.

The following sensitivity analyses were performed for the HR estimates: (i) exclusion of subjects with a diagnosis within the first year after the health screening, (ii) exclusion of subjects who reported that a close relative had T2D, MI, or stroke (these two exclusions were made to account for potential reversed causality), (iii) additional mutual adjustment for intake of other dairy or related complementary products (i.e., margarine for butter estimates, and low-fat for high-fat variants and vice versa), which was done to account for positive or negative influences from other foods when low levels of the test food were consumed.

Data processing and statistical analyses were performed with SPSS version 25 (SPSS Software, IBM, Armonk, NY, USA), SAS version 9.4 (SAS Institute, Cary, NC, USA) and SIMCA P+ version 14.0 (Umetrics, Sartorius Stedim Biotech, Umeå, Sweden). All tests were two-sided, and *p* < 0.05 was considered statistically significant.

## 3. Results

### 3.1. Dairy Intake in Northern Sweden

Time trends in dairy product intake based on the 30-year continuous screening of dietary habits, i.e., January 1986 through to December 2016, in Northern Sweden, are presented in Figure 2. Non-fermented milk and cheese consumption declined over the study period. Specifically, the intake of low- and medium-fat non-fermented milk declined over the study period, whereas the intake of high-fat non-fermented milk and butter decreased until 2007 and, thereafter, began to increase.

In the study population, 153 subjects (0.2%; 11 cases and 142 non-cases) reported null intake for total dairy intake, 5.0% reported null intake for non-fermented milk, 4.8% for fermented milk, 9.1% for butter and 2.7% for cheese (Appendix A). The reported sex, age and screening year adjusted means (SD) among consumers were 1.30 (1.05) servings per day for non-fermented milk, 0.60 (0.56) for fermented milk, 1.52 (1.41) for butter, and 0.98 (0.85) for cheese. There were no seemingly relevant (though statistically significant) differences in overall mean intakes or intakes in quartile categories between the non-cases and those in the three diagnosis groups (Appendix A). The only noticeable differences were a slightly lower intake of non-fermented milk in the non-case group compared to the three outcome groups and a slightly greater intake of butter in the stroke and MI groups compared to the T2D and non-case groups. The distributions of dairy products, non-fermented and fermented milk, butter and cheese over the study period are shown in Appendix A.

### 3.2. Cardiometabolic Events and Other Characteristics of Study Participants

Characteristics of the study participants, i.e., the 86,931 non-cases, 3692 incidents of T2D cases, 4295 incidents of myocardial infarction (MI) cases, and 3654 incidents of stroke cases, are presented in Table 1. The proportion of women was lower among MI cases (28.7%) than among the non-cases, T2D and stroke groups, and the mean age was somewhat lower in the non-case group than in the case groups. Overall, BMI indicated that more than every second individual was overweight, less than every 5th person was a smoker, and the majority of people were physically inactive in their leisure time. Several lifestyle-related characteristics differed systematically between the non-case and the case groups, including smoking, physical activity in leisure time, highest education and the proportion with university education, BMI, blood lipids and glucose levels, and blood pressure (Table 1). 

### 3.3. Impact of Selection of Reference Category

As a first step, we evaluated the impact of choosing a reference category for risk assessment in the present cohort, where virtually all participants consumed any dairy product and most also consumed non-fermented and fermented milk, butter and cheese, specifically. For simplicity, we restricted this evaluation to the basic models in T2D and MI assessments. Thus, the hazard ratios with the lowest quartile among the consumers as the reference category versus when null-consumers were chosen as the reference category were compared. The latter approach resulted in lower HRs for both T2D and MI for most dairy types (Table 2, right panel). In the former case, it was common for null consumers to have an increased risk of developing T2D or MI compared to the quartile group with the lowest intake (Table 2, left panel). These results indicated that in this study population, non-consumers appear not to be representative of the rest of the population; therefore, we decided to choose the lowest quartile of the consumers as the reference category in the Cox regressions. 

### 3.4. Non-Fermented Milk Intake and Incident of T2D, MI and Stroke

The risk of developing T2D and MI was significantly greater for men in the highest intake quartile of non-fermented milk in both the basic and lifestyle-adjusted models (HR 1.17 (95% CI 1.03, 1.34) and 1.23 (95% CI 1.10, 1.37), respectively) (Table 3). Increased risk was also seen for an incident T2D in women in the basic model (HR 1.17 (95% CI 1.01, 1.35), but this association was attenuated by lifestyle adjustment. However, the effect sizes for T2D were small, i.e., the eta squared for T2D was between 0.01 and 0.03 in sex and 10-year age group strata, and for MI it was between 0.01 and 0.123, with the highest values in the youngest age group for both men and women.

Non-fermented milk intake was unrelated to an incident stroke in both sexes.

Men and women who abstained from non-fermented milk had a significantly greater risk of developing T2D (adjusted HR 1.28 (95% CI 1.04, 1.58) and 1.34 (95% CI 1.07, 1.69), respectively) (Table 3). A significantly increased risk to stroke was also found in women with no intake of non-fermented milk (HR 1.27 (95% CI 1.01, 1.61), with a similar trend in non-consuming men.

In sensitivity analyses, the exclusion of MI cases with a diagnosis within the first year after the health screening or subjects who reported that a close relative had MI, stroke or T2D did not affect the HRs substantially.

### 3.5. Butter Intake and Incident of T2D, MI andSstroke

In men, a greater intake of butter was significantly associated with a lower HR to develop T2D by the basic model, and this remained statistically significant in the second highest quartile of butter intake after lifestyle adjustment (HR 0.80 (95% CI 0.71, 0.91)). This was not seen for women. The effect sizes were very low, i.e., the eta squared by 0.01 or slightly less in all male 10-year age groups. The risk of developing MI or stroke was not systematically associated with butter intake. Men and women who reported null intake of butter had a significantly greater risk of developing T2D in the basic model and for men after lifestyle adjustment (HR 1.25 (95% CI 1.08, 1.45)) (Table 3). Sensitivity analyses did not affect the HRs substantially.

### 3.6. Fermented Milk Intake and Incident of T2D, MI and Stroke

The basic models indicated that greater intake of fermented milk compared to the reference category was associated with a reduced risk of developing T2D, MI and stroke in men and women, but all associations were attenuated by lifestyle adjustment (Table 4). The HR in non-consumers did not differ significantly from the reference category. Sensitivity analyses did not affect the HRs substantially.

### 3.7. Cheese Intake and Incident of T2D, MI and Stroke

Similar to fermented milk intake, the basic models indicated that a greater intake of cheese tended to be associated with a lower risk of developing T2D in both men and women and a lower risk for the incidence of MI and stroke in women, but lifestyle adjustment attenuated the associations except for the incidence of stroke for women in the third quartile [HR 0.79 (0.68, 0.92)] (Table 4). Female non-consumers of cheese had a significantly increased risk of developing T2D after lifestyle adjustment [HR 1.33 (95% CI 1.00, 1.76)]. A similar tendency was seen for incident of MI and stroke for men and women who avoided cheese (Table 4). Sensitivity analyses did not affect the HRs substantially.

### 3.8. Dairy Product Fat Levels and Incident of T2D, MI and Stroke

Separate analyses for low- and high-fat variants of non-fermented milk, fermented milk, and cheese were conducted among consumers (Table 5). Consumers with the highest intakes of low-fat non-fermented milk had, compared to the reference category, an increased risk of developing T2D and having a stroke, whereas no associations were found for 3% fat non-fermented milk after lifestyle adjustment. Similarly, those with the highest intake of low-fat fermented milk and cheese were found to have an increased risk for the incidence of T2D, whereas the opposite was seen for those with the highest intake of high-fat fermented milk and cheese. The associations remained in sensitivity analyses, including when low-fat and high-fat variants were mutually adjusted.

### 3.9. Characteristics of Non-Consumers of Non-Fermented Milk

A panel of lifestyle factors and cardiometabolic risk factors were compared for the men and women, respectively, who reported null intake of non-fermented milk (Q0) with those in the lowest consumer quartile (Q1, reference category in the Cox regressions) in multivariate partial least square (PLS) modeling. As seen in Figure 3 (upper panel), most non-consuming men (Figure 3, left) and women (Figure 3, right) clustered apart from low consumers. The most influential variables for the separation are illustrated in Circos plots for men and women, respectively (Figure 3, lower panel). Non-consumers were characterized by drinking water more often, being smokers, having higher total energy intake and energy from sucrose, and higher blood sugar after a glucose load.

## 4. Discussion

The 30-year continuous screening of dietary habits and cardiometabolic risk factors in Northern Sweden enables population-based, large-scale assessments of trends in diet intake and risk assessments for incidence of disease outcomes and their risk factors. Here, we studied time trends in dairy product intake and their associations with cardiometabolic outcomes in this population. The main finding was that high intake of non-fermented milk was associated with an increased risk of developing type 2 diabetes and myocardial infarction, but the effect sizes were small.

Historically, milk intake has been high in Sweden following the evolved lactase persistence as well as suitable conditions for cattle keeping. The population has, however, periodically been exposed to influences to select low- or high-fat dairy products [2,24]. This appears to have influenced the types and amounts of dairy products that were selected in the population based on the decrease in high-fat options around the 1990s [34,35], coinciding with a unified fat-reduction health message by the Swedish authorities. This decrease was followed by a continuous increase in the intake of butter and other higher fat dairy alternatives in the last decade, which coincided with lively discussions in media on the health effects of high-fat/low-carbohydrate diets [24]. The latter consumption pattern has led to increases in the relative intake of total and saturated fatty acids and a decrease in the relative intake of carbohydrates [34]. This type of pattern was recently confirmed to be associated with an increased mortality risk [36]. For the present study population, the association between a high-fat/low-carbohydrate diet on health outcomes remains to be explored, but several independent evaluations have reported a positive association between non-fermented milk intake (a proxy for intake of saturated fat) and all-cause mortality in the Swedish population [37,38,39].

The present study is a prospective observational cohort study with strengths and weaknesses associated with such a study design. The specific strengths of this study are, in addition to the prospective design, the fairly homogenous population from a cultural and genetic ancestry perspective and the large population-based study cohort with no evidence of an influential selection bias [25]. This rendered power for sex-stratified analyses. Furthermore, compared to many other studies, dairy intake is common, and the distribution of dairy exposure is wide.

There are also limitations to acknowledge. First, dietary recordings are generally associated with both under- and over-reporting and limitations in food item specificity. This is also true for the FFQ used in this study. In the present study population, participants appear to be feasibly aware of and genuine about their dairy intake as judged from the previously reported high relative validity, i.e., comparing FFQ information with that from ten repeated 24-hour recalls [28]. Furthermore, the follow-up period was long. We do not have information on intake changes on the individual level but speculate the likelihood that some participants changed their intake pattern of dairy products over the follow-up period. This is one reason why prospective investigations on diet intake need to be based on large cohorts to compensate for loss of power from loss of exposure precision. One more potential limitation is that after the exclusion of cases and other potential reasons for reversed causality, the mean age was lower in the non-case group than in the case groups. Consequently, age was adjusted for as a continuous variable in all mean and risk assessments. One can also reflect on whether the most appropriate confounders were adjusted for. Alternatively, given that several associations were lost by lifestyle adjustment if we over-adjusted when including our set of lifestyle factors in the models. We have recently shown that dairy product intake is associated with lifestyle patterns with, e.g., a reversed causality association between BMI and elevated dairy product intakes [40]. Hence, separating true effects of dairy intake from those with its ties with other lifestyle factors is challenging and calls for identification and use of valid biomarkers or population stratification adjusted Mendelian randomization studies. One additional aspect that may be seen as a limitation is the fact that the subjects with mixed diagnoses were excluded and not analyzed separately, which might reveal interactions in situations with more than one diagnosis. However, due to power limitations in the sub-groups representing the three possible outcome combinations for T2D, MI and stroke such analyses must await further growth of the cohort. 

The risk of developing T2D, MI and stroke has been evaluated in several recent meta-analyses [10,11,16], including the recent PURE study [8]. In general, a lower risk or no association is reported for total dairy intake, non-fermented milk, fermented milk and cheese and the studied cardiometabolic outcomes. Only single studies indicate a greater risk with any of the cardiometabolic outcomes [9]. The results in the present study conform to the majority of studies in that both non-fermented milk and cheese were associated with lower HR in the basic models, but adjustment for multiple potential lifestyle confounders attenuated the effect. However, the conclusion of a positive association between high intake of non-fermented milk and increased risk of developing T2D and MI contrasts with many other studies [8,13]. One major contributing factor to the disagreement (and a difference from other published population-based cohort studies) is the selection of the reference category [8]. Thus, as illustrated in our results, the conclusion would be that the intake of non-fermented milk is protective if the null consumers were chosen as referents. The results were the same if the null-consumers and those with the lowest intake were merged into one reference group. Since the null-consumers of non-fermented milk differ in their risk and risk profile in this population, we argue that null-consumers should not form or be included in the reference group. This may be different in other populations.

Additional aspects that should be kept in mind when comparing the results of the present study with others are that in the present study, population milk and dairy product intake is both more prevalent and greater than in many other populations. Thus, only 0.2% abstained from any type of dairy product, and 5% abstained from non-fermented milk. However, the intake of non-fermented milk only explained approximately 0.2% of T2D and MI development, which is a very low effect size that is in agreement with a recent meta-analysis that reported small effect sizes for increased T2D risk with increased intake of dairy fat [41]. Thus, all the associations between milk and T2D or MI could likely be neglected as a significant risk factor for T2D and MI, even in this high-consuming population.

A noteworthy finding of the present study was that non-consumers of non-fermented milk and butter had a statistically significantly increased risk of developing T2D. Other non-consumer groups, with a few exceptions, tended to follow the same pattern for the outcomes, but most associations were not statistically significant. Hence, in this population, non-consumers of at least non-fermented milk appear to differ from consumers by more factors than avoiding milk. In addition, smoking and sugar consumption were found to be more common among non-consumers than among low consumers of milk, and it may be anticipated that general health conditions, such as gastrointestinal or autoimmune inflammatory conditions where milk avoidance has been claimed to relieve symptoms, may be more prevalent. Some of these conditions are known to be associated with an increased risk of MI [42].

Similar to many other studies, the associations between dairy exposure and health outcomes were not always consistent for the sexes [27,43]. The reason for sex deviations in the risk assessment, such as for non-fermented milk on incident of MI, can only be speculated on but may involve true biological differences that are well documented for MI and hormonal differences. A difference may also, at least in part, reflect differences in awareness of food intake and precision in the recordings per se but also differences in power conditions, e.g., for MI.

An unexpected and unexplained finding was the increased hazard ratios for high consumers of low-fat dairy products. This finding is in line with a previous study from southern Sweden [44], but differs from many other studies [13,14,15]. A possible explanation is that subjects in a well-informed population, such as the present, choose low-fat variants because they are conscious of a health risk. This would be in agreement with the association between high BMI and lower consumption of low-fat dairy variants reported for this population [40]. A more speculative hypothesis, but not substantiated in the present study, would be that low-fat food options are associated with the presence of some kind of health adverse components, food modifications or lack some protective component. For example, it has been suggested that bioactive substances associated with the intake of dairy lipids may be beneficial for cardiometabolic health and that these substances are more frequent in high-fat products [45], but further research is needed here.

## 5. Conclusions

An increased risk of developing type 2 diabetes and myocardial infarction with high intake of non-fermented milk was detected, but subjects abstaining from dairy products or choosing low-fat variants also had a greater risk in the study population, which is characterized by an evolutionarily high tolerance to milk and milk products (lactase persistence) and prevalent and high intake of dairy products. However, the effect sizes were very small, and the overall risk by milk intake is regarded as low. Future research should be directed at scrutinising mechanisms for the influence of milk and other dairy products on CVD and T2D related processes.

## Figures and Tables

**Figure 1 nutrients-11-00284-f001:**
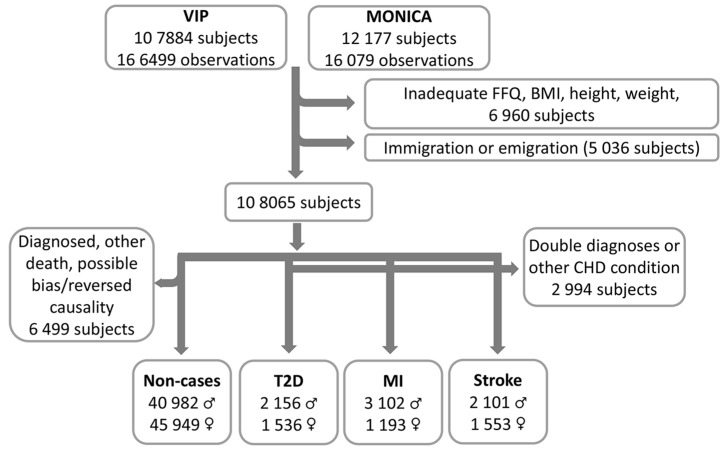
Flow chart from inclusion of the VIP and MONICA participants to the final study groups. The exclusion of potential non-cases due to possible reversed causality was for subjects who reported that a close relative had T2D, MI, or stroke.

**Figure 2 nutrients-11-00284-f002:**
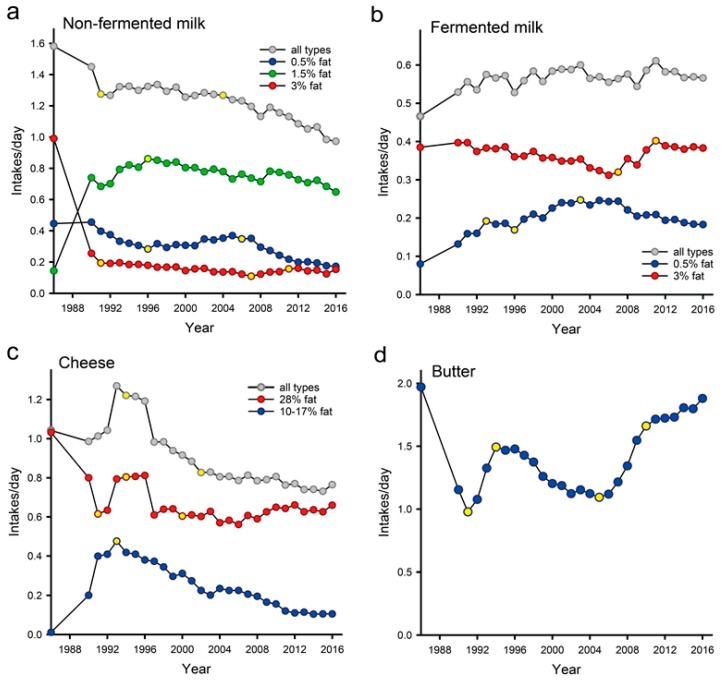
Thirty-year time trends (1986–2016) for (**a**) non-fermented, (**b**) fermented, (**c**) cheese products and (**d**) butter intake. Data are presented as the standardized means for sex and age. The yellow dots define time points were a significant trend change occurred.

**Figure 3 nutrients-11-00284-f003:**
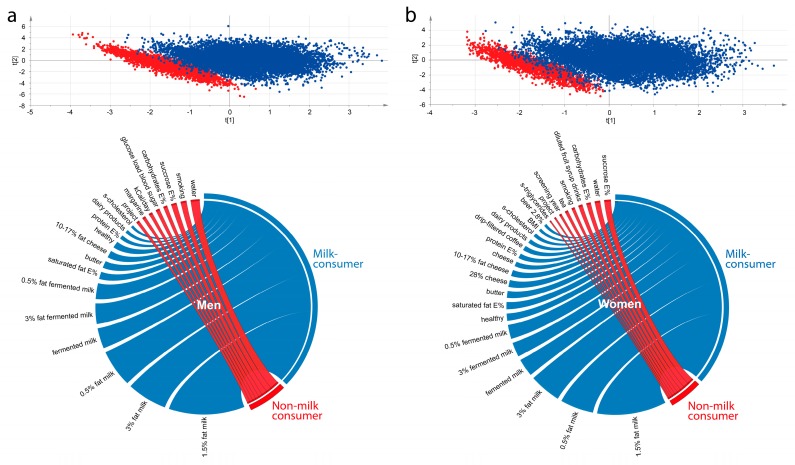
Separation of non-consumers of non-fermented milk (Q0) from (**a**) men and (**b**) women in the lowest consumption quartile (Q1, reference category in the Cox regressions) based on a panel of lifestyle and medical risk factors for cardio-metabolic diseases in partial least square (PLS) modeling for men (upper panel, left) and women (upper panel, right), respectively. The most influential variables for the separation are illustrated in Circos plots (lower panel) based on PLS loading correlation coefficients.

**Table 1 nutrients-11-00284-t001:** Baseline characteristics of the study participants. Data are presented as sex-, age- and screening-year adjusted means with standard deviations (SD), or as a percent within the group. Differences between the groups are highly significant (*p* < 0.001) for all variables. T2D for type 2 diabetes, and MI for myocardial infarction.

Characteristics	Non-Cases *n* = 86,931	T2D *n* = 3692	MI *n* = 4295	Stroke *n* = 3654
Female/male, %	52.9/47.1	41.6/58.4	27.8/72.2	42.5/57.5
Age, years (mean (SD)	45.1 (8.9)	54.2 (8.8)	54.8 (8.7)	55.7 (8.4)
University level, %	31.8	14.4	14.7	16.0
Married, %	80.9	75.7	79.2	77.5
Smoking, %				
Present	18.5	25.3	32.0	26.1
Past	28.5	33.9	32.0	31.3
Never	53.0	40.8	36.0	42.7
Swedish snus (snuff), %				
Present	18.1	16.2	17.6	13.6
Past	11.5	10.5	12.3	9.4
Never	70.4	73.2	70.1	77.0
Inactive at leisure time ^a^, %	61.4	77.4	77.0	75.2
BMI, kg/m^2^	25.7 (4.2)	29.6 (5.0)	26.6 (4.0)	26.4 (4.1)
Total cholesterol ^b^, mmol/L	5.5 (1.1)	5.5 (1.3)	5.7 (1.2)	5.5 (1.3)
Triglycerides ^b^, mmol/L	1.3 (0.8)	1.9 (1.5)	1.6 (1.1)	1.4 (1.0)
Fasting blood sugar, mmol/L	5.4 (0.8)	6.8 (2.4)	5.5 (1.3)	5.4 (1.2)
Systolic blood pressure, mmHg	125 (16)	134 (20)	128 (19)	131 (20)
Diastolic blood pressure, mmHg	78 (11)	82 (11)	80 (11)	81 (11)
Total fat intake, E%	35.5 (7.0)	36.2 (6.1)	36.2 (6.1)	36.0 (5.9)
Saturated fat intake, E%	14.8 (3.9)	15.0 (3.8)	15.3 (3.7)	15.2 (3.4)
Protein intake, E%	14.7 (2.4)	15.1 (2.4)	14.8 (2.1)	14.8 (2.3)
Carbohydrate intake, E%	47.4 (7.1)	46.7 (6.7)	47.2 (6.1)	47.1 (5.8)
Sucrose, E%	6.3 (3.5)	6.2 (4.1)	6.7 (3.9)	6.7 (3.8)

(a) Being inactive was defined as never or occasionally training in leisure time. (b) These means were also adjusted for if the analyses were done before or after 1 September 2009 when the method was changed.

**Table 2 nutrients-11-00284-t002:** Comparisons of the hazard ratios for incident T2D or MI when the lowest quartile of the consumers is taken as the reference category (as in the present study) versus when the non-consumer group is taken as the reference category. Only data from the basic models (adjusted for gender, age and screening year) are shown. Statistically significant *p*-values are in superscript.

	Low Consumption Group as Reference		Non-Consumer Group as Reference	
	T2D	MI	T2D	MI
**Non-fermented milk**				
Q0 (non-consumers)	1.37 (1.18, 1.60) ^<0.001^	1.21 (1.04, 1.40) ^0.016^	1.00	1.00
Q1 (low quartile)	1.00	1.00	0.72 (0.63, 0.85) ^<0.001 <0.001^	0.83 (0.71, 0.97) ^0.016^
Q2	1.07 (0.97, 1.17)	1.08 (0.98, 1.18)	0.78 (0.67, 0.90) ^0.001^	0.89 (0.77, 1.03)
Q3	0.96 (0.86, 1.06)	0.98 (0.89, 1.08)	0.70 (0.60, 0.81) ^<0.001^	0.81 (0.70, 0.95) ^0.008^
Q4 (high quartile)	1.15 (1.05, 1.26) ^0.003^	1.15 (1.06, 1.26) ^0.002^	0.84 (0.72, 0.97) ^0.020^	0.95 (0.82, 1.11)
**Fermented milk**				
Q0 (non-consumers)	1.16 (1.00, 1.34) ^0.047^	1.00 (0.87, 1.14)	1.00	1.00
Q1 (low quartile)	1.00	1.00	0.86 (0.75, 1.00) ^0.047^	1.00 (0.87, 1.15)
Q2	0.94 (0.86, 1.03)	0.91 (0.83, 0.99) ^0.021^	0.81 (0.70, 0.94) ^0.005^	0.91 (0.79, 1.05)
Q3	0.85 (0.78, 0.93) ^0.001^	0.83 (0.76, 0.91) ^<0.001^	0.74 (0.64, 0.85) ^<0.001^	0.84 (0.73, 0.96) ^0.012^
Q4 (high quartile)	0.82 (0.75, 0.91) ^<0.001^	0.80 (0.73, 0.87) ^<0.001^	0.71 (0.61, 0.82) ^0.020^	0.80 (0.69, 0.92) ^0.002^
**Butter**				
Q0 (non-consumers)	1.27 (1.14, 1.41) ^<0.001^	1.09 (0.98, 1.22)	1.00	1.00
Q1 (low quartile)	1.00	1.00	0.79 (0.71, 0.88) ^<0.001^	0.92 (0.82, 1.03)
Q2	0.97 (0.88, 1.07)	1.05 (0.96, 1.15)	0.77 (0.68, 0.86) ^<0.001^	0.96 (0.86, 1.08)
Q3	0.83 (0.76, 0.92) ^<0.001^	1.10 (1.01, 1.20) ^0.026^	0.66 (0.59, 0.74) ^<0.001^	1.01 (0.90, 1.13)
Q4 (high quartile)	0.88 (0.80, 0.97) ^0.011^	0.99 (0.91, 1.09)	0.70 (0.62, 0.78) ^<0.001^	0.91 (0.81, 1.02)
**Cheese**				
Q0 (non-consumers)	1.13 (0.93, 1.37)	1.20 (1.01, 1.42) ^0.042^	1.00	1.00
Q1 (low quartile)	1.00	1.00	0.88 (0.73, 1.07)	0.84 (0.70, 0.99) ^0.042^
Q2	0.91 (0.83, 1.00) ^0.043^	0.92 (0.84, 1.00)	0.80 (0.66, 0.97) ^0.026^	0.77 (0.65, 0.91) ^0.003^
Q3	0.74 (0.67, 0.81) ^<0.001^	0.88 (0.80, 0.96) ^0.003^	0.65 (0.54, 0.79) ^<0.001^	0.73 (0.61, 0.87) ^<0.012^
Q4 (high quartile)	0.83 (0.76, 0.91) ^<0.001^	0.88 (0.81, 0.96) ^0.002^	0.74 (0.61, 0.89) ^0.002^	0.73 (0.62, 0.87) ^<0.001^

**Table 3 nutrients-11-00284-t003:** Hazard ratios (95% CI) for incident type 2 diabetes (T2D), myocardial infarction (MI), or stroke by non-fermented milk or butter intake. N refers to the number of cases in each group. Statistically significant *p*-values are displayed in superscript.

		Men			Women	
	N	Basic Model	Adjusted Model	N	Basic Model	Adjusted Model
**Non-fermented milk**						
**T2D**						
Q0 (non-consumers)	114	1.39 (1.13, 1.70) ^0.002^	1.28 (1.04, 1.58) ^0.022^	95	1.36 (1.08, 1.70) ^0.009^	1.34 (1.07, 1.69) ^0.012^
Q1 (low)	428	1.00	1.00	328	1.00	1.00
Q2	565	1.14 (1.00, 1.29) ^0.044^	1.19 (1.05, 1.35) ^0.008^	397	0.98 (0.85, 1.13)	0.99 (0.86, 1.15)
Q3	440	0.98 (0.86, 1.12)	1.08 (0.95, 1.24)	287	0.94 (0.80, 1.10)	0.95 (0.81, 1.12)
Q4 (high)	609	1.14 (1.00, 1.29) ^0.043^	1.17 (1.03, 1.34) ^0.018^	429	1.17 (1.01, 1.35) ^0.031^	1.09 (0.94, 1.29)
**MI**						
Q0 (non-consumers)	139	1.21 (1.01, 1.46) ^0.039^	1.18 (0.98, 1.42)	67	1.18 (0.90, 1.55)	1.19 (0.10, 1.56)
Q1 (low)	610	1.00	1.00	262	1.00	1.00
Q2	752	1.14 (1.02, 1.27) ^0.018^	1.15 (1.03, 1.28) ^0.013^	326	0.93 (0.79, 1.10)	0.95 (0.81, 1.12)
Q3	671	1.03 (0.92, 1.15)	1.08 (0.97, 1.21)	218	0.86 (0.72, 1.03)	0.93 (0.77, 1.11)
Q4 (high)	930	1.22 (1.10, 1.35) ^<0.001^	1.23 (1.10, 1.37) ^<0.001^	320	0.99 (0.84, 1.17)	0.92 (0.77, 1.10)
**Stroke**						
Q0 (non-consumers)	95	1.21 (0.97, 1.52)	1.15 (0.92, 1.45)	92	1.29 (1.02, 1.62) ^0.031^	1.27 (1.01, 1.61) ^0.042^
Q1 (low)	413	1.00	1.00	340	1.00	1.00
Q2	542	1.14 (1.00, 1.29)	1.14 (1.00, 1.30) ^0.044^	398	0.93 (0.80, 1.07)	0.94 (0.81, 1.09)
Q3	449	1.00 (0.87, 1.14)	1.04 (0.91, 1.19)	314	0.99 (0.85, 1.16)	1.05 (0.90, 1.23)
Q4 (high)	602	1.11 (0.98, 1.26)	1.10 (0.97, 1.26)	409	1.03 (0.89, 1.19)	1.03 (0.88, 1.20)
**Butter**						
**T2D**						
Q0 (non-consumers)	261	1.33 (1.15, 1.54) ^<0.001^	1.25 (1.08, 1.45) ^0.003^	238	1.21 (1.03, 1.42) ^0.022^	1.15 (0.98, 1.35)
Q1 (low)	573	1.00	1.00	380	1.00	1.00
Q2	457	0.95 (0.84, 1.08)	1.03 (0.91, 1.17)	322	0.99 (0.82, 1.10)	1.04 (0.90, 1.21)
Q3	436	0.76 (0.67,0.86) ^<0.001^	0.80 (0.71, 0.91) ^0.001^	324	0.95 (0.82, 1.10)	0.94 (0.81, 1.09)
Q4 (high)	429	0.87 (0.77, 0.99) ^0.031^	0.96 (0.84, 1.10)	272	0.89 (0.76, 1.04)	0.90 (0.76, 1.05)
**MI**						
Q0 (non-consumers)	271	1.05 (0.91, 1.20)	0.99 (0.86, 1.14)	170	1.19 (0.99, 1.45)	1.15 (0.95, 1.40)
Q1 (low)	752	1.00	1.00	271	1.00	1.00
Q2	632	1.06 (0.95, 1.17)	1.04 (0.94, 1.16)	233	1.02 (0.86, 1.22)	1.05 (0.89, 1.26)
Q3	826	1.07 (0.97, 1.19)	1.03 (0.94, 1.14)	291	1.19 (1.01, 1.41) ^0.039^	1.12 (0.95, 1.33)
Q4 (high)	621	0.97 (0.87, 1.08)	0.99 (0.89, 1.11)	228	(0.90, 1.28)	1.04 (0.87, 1.24)
**Stroke**						
Q0 (non-consumers)	198	1.16 (0.98, 1.36)	1.10 (0.94, 1.31)	213	1.06 (0.90, 1.26)	1.04 (0.88, 1.23)
Q1 (low)	511	1.00	1.00	387	1.00	1.00
Q2	431	1.16 (1.02, 1.32) ^0.026^	1.14 (1.00, 1.30) ^0.048^	311	0.95 (0.82, 1.11)	0.96 (0.82, 1.11)
Q3	521	1.02 (0.90, 1.15)	0.99 (0.87, 1.12)	357	1.02 (0.88, 1.18)	0.97 (0.84, 1.12)
Q4 (high)	440	1.04 (0.91, 1.18)	1.02 (0.89, 1.17)	285	0.94 (0.80, 1.09)	0.92 (0.78, 1.08)

Basic models include dairy product categories (non-consumers and quartiles), sex, age and screening year. Full models were also adjusted for BMI, education, physical activity in leisure time, smoking, self-reported family history of cardiovascular disease or type 2 diabetes, screening project, and quintiles of red meat, wholegrain, fruit and vegetables and energy.

**Table 4 nutrients-11-00284-t004:** Hazard ratios for incident type 2 diabetes (T2D), myocardial infarction (MI), or stroke by intake of fermented milk or cheese. N refers to the number of cases in each group. Statistically significant *p*-values are displayed in superscript.

		Men			Women	
	N	Basic Model ^a^	Adjusted Model ^b^	N	Basic Model ^a^	Adjusted Model ^b^
**Fermented milk**						
**T2D**						
Q0 (non-consumers)	157	1.18 (0.99, 1.40)	1.16 (0.97, 1.39)	69	1.12 (0.87, 1.45)	1.21 (0.93, 1.57)
Q1 (low)	550	1.00	1.00	384	1.00	1.00
Q2	536	0.96 (0.85, 1.07)	0.98 (0.87, 1.11)	369	0.92 (0.80, 1.06)	0.99 (0.86, 1.15)
Q3	514	0.86 (0.76, 0.97) ^0.016^	0.94 (0.83, 1.06)	374	0.84 (0.73, 0.97) ^0.019^	1.00 (0.87, 1.16)
Q4 (high)	399	0.76 (0.67, 0.87) ^<0.001^	0.92 (0.80, 1.06)	340	0.90 (0.78, 1.04)	1.11 (0.95, 1.30)
**MI**						
Q0 (non-consumers)	179	0.94 (0.80, 1.10)	0.94 (0.80, 1.11)	62	1.24 (0.94, 1.63)	1.30 (0.99, 1.71)
Q1 (low)	813	1.00	1.00	318	1.00	1.00
Q2	767	0.94 (0.85, 1.04)	0.99 (0.89, 1.09)	276	0.81 (0.69, 0.95) ^0.011^	0.89 (0.76, 1.05)
Q3	754	0.87 (0.78, 0.96) ^0.004^	0.97 (0.88, 1.08)	278	0.73 (0.62, 0.86) ^<0.001^	0.86 (0.73, 1.01)
Q4 (high)	589	0.78 (0.70, 0.87) ^<0.001^	0.92 (0.82, 1.03)	259	0.83 (0.71, 0.98) ^0.029^	1.00 (0.84, 1.18)
**Stroke**						
Q0 (non-consumers)	133	1.11 (0.92, 1.35)	1.08 (0.89, 1.31)	74	1.17 (0.91, 1.50)	1.18 (0.92, 1.51)
Q1 (low)	506	1.00	1.00	391	1.00	1.00
Q2	553	1.05 (0.93, 1.19)	1.08 (0.96, 1.22)	384	0.86 (0.74, 0.99) ^0.032^	0.92 (0.80, 1.06)
Q3	523	0.96 (0.85, 1.09)	1.03 (0.91, 1.17)	414	0.94 (0.82, 1.08)	1.04 (0.90, 1.20)
Q4 (high)	386	0.82 (0.72, 0.93) ^0.003^	0.91 (0.79, 1.05)	290	0.76 (0.66, 0.89) ^<0.001^	0.87 (0.75, 1.03)
**Cheese**						
**T2D**						
Q0 (non-consumers)	63	0.97 (0.75, 1.26)	1.04 (0.79, 1.36)	56	1.40 (1.06, 1.85) ^0.019^	1.33 (1.00, 1.76) ^0.048^
Q1 (low)	556	1.00	1.00	401	1.00	1.00
Q2	534	0.93 (0.83, 1.05)	1.01 (0.89, 1.13)	328	0.87 (0.75, 1.01)	0.91 (0.79, 1.06)
Q3	420	0.76 (0.67, 086) ^<0.001^	0.89 (0.78, 1.01)	334	0.70 (0.61, 0.81) ^<0.001^	0.77 (0.66, 0.89) ^<0.001^
Q4 (high)	583	0.86 (0.77, 0.97) ^0.013^	1.00 (0.88, 1.13)	417	0.79 (0.69, 0.91) ^0.001^	0.89 (0.77, 1.04) ^3^
**MI**						
Q0 (non-consumers)	106	1.17 (0.95, 1.43)	1.19 (0.97, 1.46)	40	1.29 (0.93, 1.80)	1.27 (0.91, 1.76)
Q1 (low)	745	1.00	1.00	304	1.00	1.00
Q2	746	0.98 (0.89, 1.09)	1.04 (0.94, 1.15)	212	0.76 (0.64, 0.91) ^0.002^	0.82 (0.69, 0.98) ^0.032^
Q3	662	0.89 (0.80, 0.98) ^0.022^	0.96 (0.86, 1.07)	309	0.85 (0.73, 1.00) ^0.046^	0.94 (0.80, 1.10)
Q4 (high)	843	0.92 (0.83, 1.01)	1.03 (0.93, 1.15)	328	0.88 (0.67, 0.92) ^0.003^	0.92 (0.78, 1.09) ^3^
**Stroke**						
Q0 (non-consumers)	74	1.20 (0.94, 1.53)	1.20 (0.94, 1.54)	48	1.28 (0.95, 1.73)	1.29 (0.95, 1.75)
Q1 (low)	510	1.00	1.00	385	1.00	1.00
Q2	476	0.92 (0.81, 1.04)	0.92 (0.84, 1.08)	344	0.94 (0.81, 1.09)	0.99 (0.85, 1.15)
Q3	448	0.87 (0.77, 0.99) ^0.034^	0.93 (0.82, 1.07)	334	0.74 (0.64, 0.86) ^<0.001^	0.79 (0.68, 0.92) ^0.003^
Q4 (high)	593	0.95 (0.84, 1.07)	1.03 (0.90, 1.17)	442	0.86 (0.75, 0.98) ^0.027^	0.98 (0.85, 1.14) ^3^

(a) Basic models include dairy product categories (non-consumers and quartiles), sex, age and screening year. (b) Full models were also adjusted for BMI, education, physical activity in leisure time, smoking, self-reported family history of cardiovascular disease or type 2 diabetes, screening project, and quintiles of red meat, wholegrain, fruit and vegetables and energy.

**Table 5 nutrients-11-00284-t005:** Hazard ratios for incident T2D, MI, or stroke among consumers of low or high fat variants of non-fermented milk, fermented milk, and cheese. N refers to the number of cases in each group. Statistically significant *p*-values in superscript.

	Low Fat Variant			High Fat Variant		
	N	Basic Model	Adjusted Model	N	Basic Model	Adjusted Model
**Non-fermented milk**						
**T2D**						
Q1 (low)	564	1.00	1.00	443	1.00	1.00
Q2	287	1.07 (0.93, 1.23)	1.03 (0.89, 1.19)	273	1.12 (0.95, 1.31)	1.11 (0.94, 1.30)
Q3	425	1.12 (0.98, 1.27)	1.07 (0.94, 1.21)	382	1.06 (0.93, 1.22)	1.08 (0.94, 1.23)
Q4 (high)	625	1.39 (1.24, 1.56) ^<0.001^	1.26 (1.12, 1.41) ^<0.001^	380	1.06 (0.92, 1.22)	1.02 (0.89, 1.18)
**MI**						
Q1 (low)	670	1.00	1.00	589	1.00	1.00
Q2	311	1.04 (0.91, 1.19)	1.05 (0.91, 1.20)	316	0.99 (0.86, 1.14)	0.96 (0.83, 1.11)
Q3	407	0.96 (0.85, 1.09)	0.96 (0.84, 1.08)	475	0.99 (0.88, 1.12)	0.99 (0.88, 1.12)
Q4 (high)	621	1.10 (0.99, 1.23)	1.09 (0.97, 1.21)	564	1.10 (0.98, 1.24)	1.03 (0.91, 1.17)
**Stroke**						
Q1 (low)	554	1.00	1.00	459	1.00	1.00
Q2	254	(0.79, 1.06)	0.92 (0.79, 1.07)	216	0.89 (0.76, 1.06)	0.86 (0.72, 1.02)
Q3	371	0.96 (0.84, 1.10)	0.97 (0.84, 1.10)	419	1.13 (0.99, 1.29)	1.11 (0.96, 1.27)
Q4 (high)	510	1.18 (1.04, 1.33) ^0.008^	1.17 (1.04, 1.33) ^0.010^	501	1.26 (1.11, 1.44) ^<0.001^	1.19 (1.04, 1.37) ^0.010^
**Fermented milk**						
**T2D**						
Q1 (low)	783	1.00	1.00	1000	1.00	1.00
Q2	701	1.05 (0.95, 1.16)	1.01 (0.91, 1.12)	714	1.08 (0.98, 1.19)	1.09 (0.99, 1.21)
Q3	606	1.22 (1.09, 1.35) ^<0.001^	1.12 (1.01, 1.25) ^0.038^	881	0.92 (0.84, 1.00)	0.98 (0.89, 1.07)
Q4 (high)	661	1.16 (1.04, 1.28) ^0.007^	1.20 (1.08, 1.33) ^0.001^	623	0.76 (0.69, 0.84) ^<0.001^	0.87(0.78, 0.96) ^0.007^
**MI**						
Q1 (low)	1009	1.00	1.00	1096	1.00	1.00
Q2	667	0.96 (0.87, 1.06)	0.97 (0.88, 1.07)	867	1.11 (1.01, 1.21) ^0.028^	1.11 (1.01, 1.21) ^0.027^
Q3	720	1.01 (0.92, 1.11)	0.99 (0.90, 1.10)	1094	1.02 (0.94, 1.11)	1.09 (1.00, 1.19) ^0.045^
Q4 (high)	641	0.92 (0.84, 1.02)	0.98 (0.89, 1.09)	805	0.89 (0.81, 0.98) ^0.013^	1.02 (0.93, 1.12)
**Stroke**						
Q1 (low)	765	1.00	1.00	903	1.00	1.00
Q2	665	1.08 (0.97, 1.20)	1.10 (0.99, 1.22)	675	1.15 (1.04, 1.27) ^0.006^	1.16 (1.05, 1.28) ^0.006^
Q3	661	1.18 (1.06, 1.31) ^0.002^	1.18 (1.06, 1.31) ^0.003^	964	1.07 (0.98, 1.17)	1.11 (1.01, 1.21) ^0.036^
Q4 (high)	535	0.98 (0.88, 1.09)	1.04 (0.93, 1.16)	697	0.93 (0.84, 1.03)	1.01 (0.91, 1.12)
**Cheese**						
**T2D**						
Q1 (low)	575	1.00	1.00	816	1.00	1.00
Q2	613	1.11 (0.99, 1.24)	1.11 (0.99, 1.24)	857	0.99 (0.91, 1.09)	1.00 (0.91, 1.11)
Q3	892	1.26 (1.14, 1.40) ^<0.001^	1.21 (1.09, 1.35) ^<0.001^	568	0.85 (0.77, 0.95) ^0.004^	0.94 (0.84, 1.05)
Q4 (high)	768	1.19 (1.06, 1.33) ^0.002^	1.16 (1.04, 1.30) ^0.010^	922	0.80 (0.73, 0.88) ^<0.001^	0.93 (0.84, 1.03)
**MI**						
Q1 (low)	685	1.00	1.00	991	1.00	1.00
Q2	663	1.00 (0.90, 1.11)	1.02 (0.91, 1.13)	867	1.06 (0.97, 1.16)	1.05 (0.96, 1.15)
Q3	1021	1.14 (1.03, 1.26) ^0.009^	1.14 (1.03, 1.25) ^0.012^	611	0.92 (0.83, 1.02)	0.94 (0.85, 1.05)
Q4 (high)	869	1.06 (0.96, 1.17)	1.09 (0.98, 1.21)	1204	0.94 (0.86, 1.02)	1.02 (0.94, 1.12)
**Stroke**						
Q1 (low)	572	1.00	1.00	841	1.00	1.00
Q2	643	1.15 (1.03, 1.29) ^0.015^	1.17 (1.05, 1.31) ^0.006^	707	0.99 (0.90, 1.10)	0.99 (0.90, 1.10)
Q3	819	1.15 (1.03, 1.28) ^0.012^	1.14 (1.02, 1.27) ^0.017^	561	0.94 (0.84, 1.05)	0.97 (0.87, 1.09)
Q4 (high)	734	1.08 (0.97, 1.21)	1.11 (0.99, 1.24)	1028	0.93 (0.85, 1.02)	0.99 (0.90, 1.09)

Basic models included dairy product categories (non-consumers and quartiles), sex, age and screening year. Full models were also adjusted for BMI, education, physical activity in leisure time, smoking, self-reported family history of cardiovascular disease or type 2 diabetes, screening project, and quintiles of red meat, wholegrain, fruit and vegetables and energy.

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
