# Peer review of "Dairy Product Intake and Cardiometabolic Diseases in Northern Sweden: A 33-Year Prospective Cohort Study"

_nutrients, 2019, doi:10.3390/nu11020284_

Round 1
Reviewer 1 Report
A very interesting study, very well conducted. I have no additionnal remarks regardind this good paper.
Author Response
A very interesting study, very well conducted. I have no additional remarks regarding
A very interesting study, very well conducted. I have no additional remarks regarding this good paper.
Response: Thanks for the comment.
Reviewer 2 Report
Two versions of questionnaire were used. Any validation on the reliability and effectiveness of these two different versions? Although the questions related to dairy consumption were the same, other questions may also affect the answers of the questions targeted.
Author Response
Two versions of questionnaire were used. Any validation on the reliability and effectiveness of these two different versions? Although the questions related to dairy consumption were the same, other questions may also affect the answers of the questions targeted.
Response: As the reviewer mentions the questions on both dairy and alcohol containing products (used in the paper) were identical in all FFQ versions, and the two blocks of questions were placed in the same sections in both FFQ variants. We have previously reported that the associations between estimated intake of single nutrients and biomarkers were very similar for the long and short versions when the underlying FFQ questions were virtually identical. Thus, there is support for that assessments building on identical sections of the long and short version are valid. We have clarified this and the new text reads: “The relative validities were estimated against repeated 24-hour dietary records and biological markers, including a comparison between the long and short version”. The basis is found in reference 29.
Reviewer 3 Report
Although the relationship between dairy products and cardiometabolic diseases is not new, the authors have made a significant contribution to the literature by the use of rigorously assessing this relationship, and findings are important for public health.
My key comment is related to the grouping of subjects due to single outcomes only. In my opinion, the next study group should be considered – containing subjects who were diagnosed with two or more outcomes for showing mixed health outcomes also in this group.
Title: I suggest completing the title by adding ‘intake’ and years of observation. |
Lines 29-30: Point (name) which ‘low-fat variants of the targeted dairy products had increased risk for T2D, MI, or stroke’ – now, it is not clear. |
Line 32: The phrase ‘a population characterized by high milk tolerance’ is not supported by the results of the research and should be appear in introduction section or the first sentence of abstract instead of conclusion of the abstract. |
Lines 34-36: The last phrase of the abstract is not related (is not justified by the results) to the results given previously in the abstract body |
Introduction in general: There is no explanation why the study was focused on type 2 diabetes, myocardial infarction and stroke? |
Line 43: The phrase ‘the common denominator for dairy products’ is unfortunate. |
Lines 43-45: I don’t agree that milk spożywcze not contains ‘probiotic bacteria’ e.g. due to a pasteurisation process. |
Lines 65-71: It is not clear why this paragraph is focused on low genetic variation of Swedish population while the study not aimed on this area. Instead of this, the Introduction section should show more details regarding inconsistency mentioned in lines 55-57. |
Methods section, in general: There is no detailed explanation regarding dairy foods. What it means ‘low- and medium-fat non-fermented milk’ etc. Are Swidish consumers aware about fat content (e.g. in cheese) to give adequate/proper answers? What about mixed spreads containing butter? Morever, how servings od dairy foods were calculated? |
Lines 95-99: How many subjects were excluded due to reasons given in the sentence? The Figure 1 contains only chosen or agregated data in regards to excluded subjects. Futrhetmore, what was the reason to exclude subjects with height <130 or >210 cm or weight <35 kg (instead of other cut-off, e.g. <36 kg)? |
Figure 1: How possible bias/reversed causality was measured? This explanation is put into 194-195 lines, but also should be given as notes to the figure. |
Line 114: It is confusing to read that FFQ consists of 64-66 food items. How many items exactely: 64 or 65 or 66? |
Line 115: How many questions was related to dairy products? How dairy products were agregated/divided into food goups when questionning? |
Lines 118-121: What about dairy products? How meal-time portion sizes were presented for this type of foods? |
Lines 141-142: The more appropriate an abbreviation for total cholesterol is e.g. ‘TChol’ instead of ‘S-cholesterol’, for - triglycerides – ‘TG’ instead of ‘S-triglycerides ‘. |
Lines 153-157: Please, mention all categories of confounders such as education, physical activity and foods and energy intake (put cut-offs or ranges). |
Lines 161-162: Please, put ranges for quartiles of dairy food items. |
Lines 185-187: What was the result of the analysis when as the reference category were considered ‘non-consumers’? |
Fig.2: Standardized means of intakes/day are related to amount in grams or servings? |
Tab.4: Not complete data regarding some ranges of 95%CI |
Disscusion, in general: Well done, with many speculations properly supported by citations, with one exception. Furthermore, it is worth to discuss a limitation of the study due to the exclusion of ‘subjects with more than one outcome diagnosis at the first event were excluded from this study’. |
Lines 394-396: The statement should be supported by citations in this sentence as well as the whole paragraph. |
Lines 441-442: The phrase ‘population characterized by an evolutionarily high tolerance to milk and milk products (lactase persistence)’ is not supported by the results of the research |
Lines 446-448: The sentence appeared as ‘a ghost’ – it was not discuss in the text previously. |
Author Response
We want to thank the reviewer for constructive comments and have responded point-by-point to each of the.
Reviwer 3
Although the relationship between dairy products and cardiometabolic diseases is not new, the authors have made a significant contribution to the literature by the use of rigorously assessing this relationship, and findings are important for public health.
My key comment is related to the grouping of subjects due to single outcomes only. In my opinion, the next study group should be considered – containing subjects who were diagnosed with two or more outcomes for showing mixed health outcomes also in this group.
Response: We acknowledge that this is a way to do the grouping but did not choose to take that route due to that the power in the sub-groups representing the three possible outcome combinations was far too low, and the subjects with mixed diagnoses could neither be analyzed together nor be allocated into several main outcome groups. We therefore came to a final conclusion, which we wish to keep, that clean groups were preferential for the present research questions and also in consideration of that the data basis was large enough for this harder filtering of cases. We acknowledge that our conclusions can only be extrapolated to individuals with single outcomes.
Title: I suggest completing the title by adding ‘intake’ and years of observation.
Response: We have changed the title as suggested. The new title reads “Dairy product intake and cardiometabolic diseases in the northern Sweden population - a 33-year prospective cohort study.”
Lines 29-30: Point (name) which ‘low-fat variants of the targeted dairy products had increased risk for T2D, MI, or stroke’ – now, it is not clear.
Response: There is a word limit of a maximum of around 200 words for the journal and we have 218 words in the present version. For this reason, we do not see that there is space for further descriptions of single associations.
Line 32: The phrase ‘a population characterized by high milk tolerance’ is not supported by the results of the research and should be appear in introduction section or the first sentence of abstract instead of conclusion of the abstract.
Response: As suggested, the information was moved to the first paragraph.
Lines 34-36: The last phrase of the abstract is not related (is not justified by the results) to the results given previously in the abstract body.
Response We have clarified at the end of the results paragraph that effect-sizes were small to underpin the associated statement in the conclusion. The added text reads “Generally, effect-sizes were small”.
Introduction in general: There is no explanation why the study was focused on type 2 diabetes, myocardial infarction and stroke?
Response: We have edited the last paragraph in the introduction to clarify that incident cardiometabolic events were the study target. The new text reads “The aims of the present longitudinal cohort study were to characterize dairy food intake and its prospective association with incident cardiometabolic events, that is type 2 diabetes (T2D), myocardial infarction (MI) and stroke, in a large population with an evolutionarily high tolerance to milk”.
Line 43: The phrase ‘the common denominator for dairy products’ is unfortunate.
Response: We have changed the text and it now reads “Milk, the basis for dairy products, …”.
Lines 43-45: I don’t agree that milk spożywcze not contains ‘probiotic bacteria’ e.g. due to a pasteurisation process.
Response: We agree that the text was misleading as the example was non-fermented milk and we have deleted the words “probiotic bacteria”.
Lines 65-71: It is not clear why this paragraph is focused on low genetic variation of Swedish population while the study not aimed on this area. Instead of this, the Introduction section should show more details regarding inconsistency mentioned in lines 55-57.
Response: The paragraph intends to give a basic information on the population where the study is conducted and one phenotypical characteristic of this population that makes it stand out from most other study cohorts is the large portion of lactose tolerant subjects. We would prefer to leave the section as is.
We acknowledge the comment on presenting more detailed information on the complexity in the results from different studies but abstained from developing the section as this would require a lengthy expansion with comparisons of e.g., details on study populations and methods to be meaningful, and therefore choose to refer to review articles on the topic.
Methods section, in general:
There is no detailed explanation regarding dairy foods. What it means ‘low- and medium-fat non-fermented milk’ etc.
Response: Thanks for this comment. We have clarified this and the new text reads “The questions on dairy products, i.e., 3 questions for non-fermented milk (0.5%, 1.5% and 3.0% fat, respectively, referred to as low, medium and high fat milk), 2 questions for fermented milk (0.5 and ³3% fat, respectively, referred to as low and high variants), 3 questions for butter (pure butter on bread or for cooking and one on a mixed spread with 70% butter), and 2 questions for cheese (10-15% fat and 28 or more % fat, and referred to as low or high variants), remained unchanged over the study period.
Are Swedish consumers aware about fat content (e.g. in cheese) to give adequate/proper answers?
Response: Yes, it is mandatory to indicate the fat content on all dairy products, including cheese, and there is significant advertising and other types of information on high versus low options so, though no scientifically substantiated, it is our opinion that consumers know what fat level they consume of cheese as well as other dairy products.
What about mixed spreads containing butter?
Response: There is one mixed spread (named Bregott) on the Swedish market which dominates consumption of spreads. About 70% of the fat content in Bregott is milk fat from cream and the remaining fat is from a vegetable oil. Bregott was counted as a butter product. We have clarified this in the section where the questions of dairy products are described. Please see text above.
Morever, how servings of dairy foods were calculated?
Response: Here servings are equal to reported intakes per day.
Lines 95-99: How many subjects were excluded due to reasons given in the sentence? The Figure 1 contains only chosen or aggregated data in regards to excluded subjects. Furthermore, what was the reason to exclude subjects with height <130 or >210 cm or weight <35 kg (instead of other cut-off, e.g. <36 kg)?
Response: Information on the numbers of subjects excluded for the reasons listed on previous lines 95-99, i.e., 6,960 subjects, is presented in the Flow chart (Fig. 1). The figure also shows the numbers excluded for the reason in the next sentence, i.e., immigration or emigration. We have added this information on the numbers per se to the text too.
The cut-off limits for the anthropometric measures have been defined as physiologically implausible by the medical experts in the steering group at the Västerbotten Intervention project. We do not have the further underlaying information but follow their recommendations.
Figure 1: How possible bias/reversed causality was measured? This explanation is put into 194-195 lines, but also should be given as notes to the figure.
Response: We have added the information to the figure legend of Fig. 1. The text reads “Exclusion of potential non-cases due to possible reversed causality were for subjects who reported that a close relative had T2D, MI, or stroke”.
Line 114: It is confusing to read that FFQ consists of 64-66 food items. How many items exactely: 64 or 65 or 66?
Response: We agree, but we had no possibility to influence this FFQ modification of an expansion with two questions over time, i.e., one question on water and one on egg. Both were placed at the end of the FFQ. The two new questions were not included in estimation of energy or nutrients, i.e. all calculations followed the list in the 64-question variant. We have added information on the proportions responding to each variant. The added text reads “Among the shorter FFQ versions, 15% had responded to the 64-question variant, 20% the 65-questions variant, and 65% the 66-question variant”.
Line 115: How many questions was related to dairy products? How dairy products were aggregated/divided into food groups when questioning?
Response: We have added information on number of “dairy” questions and how they were aggregated. The edited text reads as described above.
Lines 118-121: What about dairy products? How meal-time portion sizes were presented for this type of foods?
Response: We have edited the text to clarify that the weighting for dairy products were sex and age standardized portions based on information from the validation study described in reference 28. The edited text reads “and other foods, such as dairy products, had sex- and age-specific portions sizes, or fixed sizes, e.g. an apple or egg”.
Lines 141-142: The more appropriate an abbreviation for total cholesterol is e.g. ‘TChol’ instead of ‘S-cholesterol’, for - triglycerides – ‘TG’ instead of ‘S-triglycerides ‘.
Response: We have deleted the abbreviations since they were never used later in the manuscript, except for in Table 1. Table 1 was edited to harmonize with the change.
Lines 153-157: Please, mention all categories of confounders such as education, physical activity and foods and energy intake (put cut-offs or ranges).
Response: We acknowledge the comment from the reviewer but would prefer to leave the information at its present level where information is given on the number of categories for each confounder. Our argument is that we favour not to expand the text so extensively as it would be in the case if a listing of all category options with ranges for each confounder. This expansion would be even longer if cut-off limits should be given since that information has to be given in sex and age strata to follow how ranking was done.
Lines 161-162: Please, put ranges for quartiles of dairy food items.
Response: As requested, we have added the ranges within each consumer quartile for non-fermented milk, fermented milk, butter and cheese in Table S1.
Lines 185-187: What was the result of the analysis when as the reference category were considered ‘non-consumers’?
Response: We interpret the comment as a request for the results when non-consumers are selected as the reference category. These results are shown in Table 2 for T2D and MI.
Fig.2: Standardized means of intakes/day are related to amount in grams or servings?
Response: Here we used servings per day and which is directly correlated to estimated grams per day.
Tab.4: Not complete data regarding some ranges of 95%CI
Response: We could not find a missing 95%CI per se in the tables but we found a formatting error for cheese intake and MI (women) and Q4 which was corrected.
Discussion, in general: Well done, with many speculations properly supported by citations, with one exception.
Furthermore, it is worth to discuss a limitation of the study due to the exclusion of ‘subjects with more than one outcome diagnosis at the first event were excluded from this study’.
Response: In line with our response to the reviewer´s first comment we have added the following text to the discussion: “One additional aspect that may be seen as a limitation is the fact that subjects with mixed diagnoses were excluded and not analysed separately, which might reveal interactions in situations with more than one diagnosis. However, due to power limitations in the sub-groups representing the three possible outcome combinations for T2D, MI and stroke such analyses must await further growth of the cohort”.
Lines 394-396: The statement should be supported by citations in this sentence as well as the whole paragraph.
Response: We have added references to support the statements in this section.
Lines 441-442: The phrase ‘population characterized by an evolutionarily high tolerance to milk and milk products (lactase persistence)’ is not supported by the results of the research
Response: We have rephrased the text. It now reads: “An increased risk of developing type 2 diabetes and myocardial infarction with high intake of non-fermented milk was detected, but subjects abstaining from dairy products or choosing low-fat variants also had a greater risk in the study population, which is characterized by an evolutionarily high tolerance to milk and milk products (lactase persistence) and prevalent and high intake of dairy products”
Lines 446-448: The sentence appeared as ‘a ghost’ – it was not discuss in the text previously
Response: We agree with the reviewer and have edited the sentence to read “Future research should be directed at scrutinising mechanisms for milk and other dairy products on CVD and T2D related processes.”
